# Comparative Analysis of RT-PCR and a Colloidal Gold Immunochromatographic Assay for SARS-CoV-2 Detection

**DOI:** 10.3390/diagnostics15111362

**Published:** 2025-05-28

**Authors:** Hui Li, Dakai Liu, Qiang Zhou, George D. Rodriguez, Harlan Pietz, Vishnu Singh, Eric Konadu, Keither K. James, Calvin Lui, Mingyu Shao, Junyu Chen, Andrew Schreiner, Carl Urban, James Truong, Nishant Prasad, William Harry Rodgers

**Affiliations:** 1Department of Pathology & Clinical Laboratories, NewYork-Presbyterian Queens, 56-45 Main Street Flushing, New York, NY 11355, USA; 2Clinical Laboratory, The Second Affiliated Hospital of Guangzhou Medical University, 250 Changgang East Road Zhuhai District, Guangzhou 510260, China; 3Department of Quality, Patient Safety and Regulatory Affairs, NewYork-Presbyterian Queens, 56-45 Main Street Flushing, New York, NY 11355, USA; 4Weil Cornell Medical College, 1300 York Avenue, New York, NY 10065, USA; 5Division of Infectious Disease, NewYork-Presbyterian Queens, 56-45 Main Street Flushing, New York, NY 11355, USA; 6Department of Pathology & Laboratory Medicine, Weil Cornell Medical College, 1300 York Avenue, New York, NY 10065, USA

**Keywords:** SARS-CoV-2, COVID-19, reverse transcription polymerase chain reaction, colloidal gold immunochromatographic assay, cycle threshold, point of care

## Abstract

**Background/Objectives:** The COVID-19 pandemic has highlighted the urgent need for rapid, accurate, and accessible diagnostic testing to effectively manage and contain the spread of SARS-CoV-2. RT-PCR is widely recognized as the gold standard for SARS-CoV-2 detection due to its high sensitivity and specificity. However, RT-PCR testing requires specialized laboratory equipment, highly trained personnel, and extended processing times, which limits its feasibility for large-scale screening and point-of-care applications. This study aims to systematically evaluate the diagnostic performance of RT-PCR and a colloidal gold immunochromatographic assay (GICA). **Methods**: By comparing these two methods, we seek to determine a GICA’s effectiveness as a complementary or alternative diagnostic tool, particularly in resource-limited settings and scenarios requiring rapid, large-scale testing. We assessed the following key clinical parameters: sensitivity, specificity, NPV, PPV, and accuracy. Additionally, we investigated the correlation between GICA signal intensity and RT-PCR Ct values using regression analysis, receiver operating characteristic curve analysis, and the calculated area under the curve. **Results**: Our findings indicate that while RT-PCR exhibits superior sensitivity, GICA results demonstrate a strong correlation with RT-PCR results and provide a rapid, cost-effective alternative for SARS-CoV-2 detection. Unlike RT-PCR, which requires extensive resources and prolonged turnaround times, a GICA delivers results within 20 min, making it a viable option for decentralized testing and real-time public health interventions. **Conclusions**: These results suggest that a GICA can serve as a complementary diagnostic tool alongside RT-PCR, particularly in resource-limited settings and high-throughput screening scenarios. By integrating GICAs into broader testing strategies, healthcare systems can enhance early detection efforts, improve accessibility to diagnostics, and strengthen pandemic response measures.

## 1. Introduction

Since the emergence of coronavirus disease 2019 (COVID-19) in late December 2019, the global impact of the causative agent, severe acute respiratory syndrome coronavirus 2 (SARS-CoV-2), has been profound and far-reaching. As of 14 April 2025, there have been over 777 million confirmed cases of COVID-19 worldwide, with more than 7 million deaths directly attributed to the disease (https://covid19.who.int, accessed on 11 April 2025). This unprecedented global health crisis has disrupted nearly every facet of human life, affecting not only healthcare systems but also economies, education, social structures, and daily routines. The pandemic has underscored the critical importance of resilient public health infrastructures, timely interventions, and international cooperation in addressing large-scale health emergencies. Healthcare systems across the globe have been pushed to their limits, with hospitals and clinics experiencing surges in patient volume that have overwhelmed available resources. These surges led to widespread shortages of essential medical supplies, healthcare personnel, intensive care unit (ICU) beds, and ventilators, particularly during the peak waves of infection. In particular, the need for rapid and accurate diagnostic testing emerged as a cornerstone of effective outbreak management [1,2,3,4].

Effective testing strategies have been fundamental to the early detection of COVID-19 cases and the development of appropriate treatment protocols. The widespread availability and accessibility of diagnostic tests have had a transformative impact on public health efforts worldwide. Comprehensive testing has allowed for more accurate monitoring of the virus’s spread, facilitated timely containment measures, and informed data-driven policy decisions. In doing so, diagnostic testing has played a central role in shaping targeted public health interventions aimed at reducing transmission and mitigating the broader impact of the pandemic.

Building upon the foundational role of diagnostic testing, the data generated have proven indispensable in modeling the trajectory of the pandemic, helping scientists and policymakers forecast infection trends, allocating healthcare resources, and planning vaccination campaigns more effectively [5,6]. However, the unprecedented global demand for testing, particularly during peak waves of infection, introduced significant challenges. These included disruptions in the supply chain for critical materials, shortages of trained personnel, limited laboratory capacity, and the need for highly specialized equipment.

Nevertheless, these challenges spurred significant innovation in diagnostic technologies. Traditional testing methods such as RT-PCR were supplemented by the development of faster and more accessible alternatives, including antigen-based rapid tests, point-of-care diagnostics, and at-home testing kits. These advancements significantly improved the scalability and speed of testing, enabling broader community surveillance and quicker isolation of infected individuals. As the pandemic progressed, the demand for diagnostic tools that were not only accurate and rapid but also cost-effective and easily deployable across diverse settings grew substantially. This underscored the urgent need for comprehensive evaluation and validation of emerging testing methods to ensure their reliability and effectiveness [7,8,9,10,11].

Among the various diagnostic techniques, reverse transcription polymerase chain reaction (RT-PCR) is considered the gold standard for SARS-CoV-2 detection due to its superior sensitivity and specificity [12,13]. By identifying viral RNA within a sample, RT-PCR ensures a highly reliable confirmation of infection. However, despite its accuracy, RT-PCR requires specialized laboratory equipment, trained personnel, and prolonged processing times. Furthermore, these limitations create a need for alternative diagnostic methods that complement RT-PCR in resource-constrained settings or situations requiring immediate results [14,15].

In contrast, a colloidal gold immunochromatographic assay (GICA), a rapid antigen test that provides a practical and efficient alternative. A GICA detects viral antigens, notably the nucleocapsid (N) protein, allowing for a faster and more accessible diagnostic approach. The simplicity, affordability, and ability to be deployed in non-laboratory environments make a GICA a valuable tool for widespread screening. Results are available within minutes, enabling real-time decision-making in clinical and community settings. However, like other antigen-based tests, a GICA generally exhibits lower sensitivity compared to RT-PCR, potentially leading to higher false-negative rates, especially in cases with low viral loads [16].

Given the distinct advantages and limitations of RT-PCR and a GICA, a systematic comparative analysis is essential to assess their correlation and diagnostic parameters, including sensitivity, specificity, accuracy, positive predictive value (PPV), and negative predictive value (NPV) [17]. Such an analysis is critical for optimizing testing strategies, ensuring effective resource allocation, and developing adaptable diagnostic frameworks suitable for various epidemiological, clinical, and public health contexts [18,19].

This study systematically compares the performance of the GICA and RT-PCR tests to provide empirical data on the GICA’s diagnostic capabilities. The findings aim to inform the integration of a GICA into a comprehensive SARS-CoV-2 testing strategy, offering public health officials and healthcare providers valuable insights into selecting appropriate diagnostic approaches. By refining testing strategies, this study seeks to enhance the effectiveness of responses to both current and future pandemics.

## 2. Materials and Methods

### 2.1. Sample Collection

During the study period from 15 March 2020 to 23 July 2021, nasopharyngeal swab specimens were collected and processed at our clinical laboratory for SARS-CoV-2 testing. A subset of 129 nasopharyngeal swab specimens were chosen from unvaccinated individuals with previous cycle threshold (Ct) values from RT-PCR analyses [20] to perform both real-time RT-PCR and GICA tests.

The collection methodology was standardized to ensure consistency and reliability across all samples. Nasopharyngeal swabs, the primary method employed, are recognized for their efficiency in collecting sufficient viral material for diagnostic purposes.

Each specimen was collected adhering to strict safety and hygiene protocols to protect medical personnel and patients from potential virus transmission. After collection and RT-PCR testing, remaining samples were immediately transported to the laboratory and preserved at −80 °C to maintain the integrity of the samples, ensuring that the subsequent analysis would provide reliable and valid results.

### 2.2. Molecular RT-PCR Procedure

SARS-CoV-2 RNA present in nasopharyngeal swab specimens was assayed by mRT-PCR using the Cepheid^®^ Xpert Xpress SARS-CoV-2 assay on Infinity (Cepheid Inc., Sunnyvale, CA, USA). This assay consists of two amplicons with specific sets of primers/probes. Amplicon 1 targets the region in the viral nucleocapsid gene unique to SARS-CoV-2. Amplicon 2 targets a conserved region of the viral protein envelope gene homologous to all coronaviruses of the Sarbecovirus sub-genus. In addition, a sample processing control and a probe check control are also included for the assay performance. The assay analytical sensitivity was determined by serial dilutions of ZeptoMetrix virus stock—NATSARS(CoV2)-ERC—with a known concentration. The limit of detection is 30 virions per assay [21,22].

Viral loads are quantified using Ct values, which indicate the number of amplification cycles needed for the fluorescent signal of the target RNA to exceed a detectable threshold. A lower Ct value indicates a higher viral RNA concentration, suggesting a significant viral load. Conversely, a higher Ct value implies a lower viral RNA amount, indicating a smaller viral load or less prevalent virus in the sample [23,24]. For diagnostic purposes, a Ct value above 45 was considered negative in our study based on manufacturer instruction for use.

### 2.3. Colloidal Gold Immunochromatographic Assay (GICA)

The 2019-nCoV Antigen Kit (Guangdong Hecin Scientific Inc, Guangzhou, Guangdong, P. R. China) utilizes a colloidal gold immunochromatographic assay (GICA) for rapid detection of SARS-CoV-2 antigens in nasopharyngeal swab specimens. The assay employs a membrane-based lateral flow technique. In this process, a sample is applied to a test strip embedded with a colloidal gold-labeled monoclonal antibody specific to the virus nucleocapsid (N) protein. When a sample contains SARS-CoV-2 antigens, these bind to the gold-labeled antibody. This complex migrates along the strip via capillary action to a test line where a second fixed antibody captures the complex. This captured complex forms a visible pink/purple line, indicating a positive result if the antigen is present. A separate control line (C line) with a different antigen confirms the test functionality.

The intensity of the reaction lines categorizes antigen levels into five distinct grades from 0 (negative) to 5 (+++++) as follows:

0: Only the control line is visible; no S or N lines.

1: A faint but visible S or N line.

2: S and/or N line intensity is less than 50% of the control line intensity.

3: S and/or N line intensity is more than 50% but less than 100% of the control line intensity.

4: S and/or N line intensity is closely approximating that of the control line.

5: S and/or N line intensity exceeds the control line intensity.

All procedures were meticulously performed following the manufacturer’s guidelines [25,26,27,28] (Figure 1 and Table 1).

### 2.4. Statistical Analysis

Statistical analysis considered sensitivity, specificity, positive predictive value (PPV), negative predictive value (NPV), and accuracy, which were computed using RT-PCR as the reference standard by statistical formulas. Wilson score intervals were used to calculate the 95% confidence intervals (CI) for sensitivity, specificity, PPV, NPV, and accuracy [17,18]. This method is appropriate for small sample sizes and proportions near zero or one. To examine the relationship between RT-PCR Ct values and GICA test readings, Pearson’s correlation coefficient was calculated. The Pearson correlation coefficient (r) measures the strength and direction of the linear relationship between two variables, with an r value close to -1 indicating a strong negative correlation. The receiver operating characteristic (ROC) curve [29,30] was generated to evaluate the performance of the GICA method in comparison to the RT-PCR gold standard. The area under the curve (AUC) was calculated to quantify the overall ability of the GICA method to discriminate between positive and negative cases [31].

## 3. Results

### 3.1. Diagnostic Performance of the GICA Test Compared to RT-PCR

The results of the GICA test were evaluated against the RT-PCR results to determine its diagnostic parameters (Table 2 and Table 3). The GICA test was found to be 71.8% sensitive among all RT-PCR positive samples tested. This suggests that while the GICA test is somewhat effective in detecting true positive cases, there is a notable proportion of false negatives.

The specificity of the GICA test was found to be 100%, indicating that it accurately identified all RT-PCR negative individuals. This high specificity indicates that the GICA test is highly reliable in confirming the absence of virus when the RT-PCR test result is negative.

The positive predictive value (PPV) of the GICA test was 100%, signifying that all individuals who tested positive with the GICA test were truly positive according to the RT-PCR results. This high PPV underscores the test effectiveness in correctly identifying true positive cases, minimizing the occurrence of false positives. However, the negative predictive value (NPV) of the GICA test was 26.7%, indicating that only 26.7% of the individuals who tested negative with the GICA test were truly negative according to the RT-PCR results. This relatively low NPV highlights a significant limitation of the GICA test in correctly identifying true negative cases, leading to a higher rate of false negatives.

Overall, the accuracy of the GICA test was calculated to be 74.4%, reflecting the proportion of true results (both true positives and true negatives) among the total number of cases examined. While the GICA test shows excellent specificity and PPV, its moderate sensitivity and low NPV suggest that it should be used cautiously.

### 3.2. The Correlation Between the RT-PCR and GICA Results

The relationship between RT-PCR cycle threshold (Ct) values and GICA test readings was analyzed (Figure 2). The scatter plot demonstrates a clear negative correlation between the two variables. Higher viral loads, indicated by lower Ct values, are associated with higher GICA test readings, whereas lower viral loads, reflected by higher Ct values, correspond to lower GICA test readings.

When Ct values are below 20.9, GICA readings remain consistently high, four or five, indicating robust antigen detection by the GICA test in the presence of high viral loads (Table 4). Rapid detection of the virus antigen by a GICA when viral load is high is crucial for early and accurate infection detection. As Ct values increase from 21.1 to 28.1, a noticeable decline in GICA readings is observed, with readings dropping to approximately two or three, signifying a reduction in the antigen detection capability of the GICA test as viral loads decrease.

Beyond a Ct value of 28.3, GICA readings continue to decrease. By the time the Ct value reaches 32.1, GICA readings approach 0, indicating minimal antigen detection. This trend continues for Ct values up to 44.6, where GICA readings remain near zero, reflecting the test’s limited sensitivity at very low viral loads. At a Ct value of 31.9, the GICA test reaches its sensitivity limits, which is expected for an immunoassay. Noteworthily, all 12 RT-PCR negative specimens are negative by the GICA method.

These data indicate that while the GICA test is highly effective at detecting significant virus presence, its sensitivity diminishes considerably at higher Ct values, corresponding to lower viral loads. This pattern is crucial for understanding the diagnostic utility of the GICA test. It highlights the test reliability in scenarios of high viral load, making it a valuable tool for early detection. However, the reduced sensitivity at low viral loads must be considered in clinical and public health contexts, particularly in situations requiring the detection of low levels of the virus, such as in asymptomatic or recovering patients.

The correlation coefficient (Pearson’s r) between the Ct value and GICA reading is approximately −0.95. This indicates a strong negative correlation between the two variables, suggesting that higher Ct values are associated with lower GICA results (Figure 3).

### 3.3. Evaluation of Overall GICA Method Performance Against RT-PCR

The performance of the GICA method was evaluated against the gold standard RT-PCR using the receiver operating characteristic (ROC) curve and the area under the curve (AUC). The GICA results were converted into a binary classification, where non-zero values were considered positive. The ROC curve was plotted by comparing the true positive rate (sensitivity) and the false positive rate (1 – specificity) across various threshold settings.

The AUC was calculated to be approximately 0.859, indicating that the GICA method has a fairly acceptable discriminatory ability to distinguish between positive and negative cases as defined by RT-PCR. An AUC value of 0.859 suggests that there is an 85.9% chance that the GICA method will correctly differentiate between a randomly chosen positive case and a randomly chosen negative case. This performance demonstrates that the GICA method has achieved its effectiveness as a diagnostic tool when compared to the RT-PCR gold standard (Figure 4).

## 4. Discussion

Improvements in the diagnosis of SARS-CoV-2 using simple, rapid, and cost-effective methods remain crucial for effectively managing current and potential future waves of the pandemic, both at the individual patient level and in broader public health contexts. A diverse array of rapid and economical diagnostic assays is now available, including serological tests [32,33], electrochemical immunosensors [34], and antigen detection tests [35,36]. However, comprehensive comparative studies evaluating the performance, limitations, and contextual suitability of these diagnostic approaches remain limited, highlighting a pressing need for evidence-based assessments.

This study focuses on evaluating a gold immunochromatographic assay (GICA) method for SARS-CoV-2 detection by examining its analytical performance parameters. The findings reveal a strong negative correlation between the GICA results and RT-PCR cycle threshold (Ct) values, with a Pearson’s correlation coefficient of approximately –0.95. This significant inverse relationship suggests that higher Ct values—indicative of lower viral loads—are associated with reduced GICA signal intensity. This finding underscores the GICA assay’s potential sensitivity to viral load dynamics and its utility in identifying patients with high transmissibility risk.

The GICA method demonstrated a sensitivity of 71.8%, indicating that while it may not serve as a complete substitute for RT-PCR, it still provides valuable diagnostic information rapidly. Notably, the finding that the GICA test demonstrated 100% specificity and 100% positive predictive value (PPV) is highly significant in the context of diagnostic accuracy. These metrics provide critical insights into the reliability of the test, particularly in identifying true positive cases. High specificity eliminates the possibility of false positives, which is crucial in clinical settings to avoid unnecessary anxiety, treatment, or isolation. The 100% PPV value instills high confidence in the test’s positive results, and a positive GICA result can be acted upon immediately, potentially expediting isolation, treatment, or further confirmatory testing. However, the negative predictive value (NPV) was relatively low at 26.7%, suggesting that a negative GICA result alone is not sufficiently reliable to exclude infection. The overall diagnostic accuracy of the assay was calculated at 74.4%, positioning GICA as a viable, though limited, screening tool.

These results underscore the potential utility of the GICA method as a complementary diagnostic tool. While its lower sensitivity and NPV limit its use as a standalone test, its high specificity and PPV support its application in settings where rapid identification of high viral load cases is necessary. A strategic approach, where a GICA is used as an initial screening tool with reflex RT-PCR testing for negative results, could reduce the overall burden on RT-PCR resources and lower diagnostic costs. This protocol may be particularly beneficial in evaluating symptomatic individuals where viral infection is part of the differential diagnosis. In contrast, routine screening of asymptomatic individuals may not benefit from this dual-testing strategy due to the low NPV of the GICA method.

In pandemics such as COVID-19, where asymptomatic transmission played a significant role, identifying these carriers was essential for effective control strategies. Asymptomatic individuals do not show symptoms but can still transmit the disease. Without identification, they are unlikely to isolate and may unknowingly infect vulnerable populations. Detecting asymptomatic cases enables early isolation and, where applicable, treatment, thereby reducing the window of transmission and helping to limit the spread within the community. Understanding the number and distribution of asymptomatic carriers also provides critical insights into transmission dynamics, such as the reproduction number and herd immunity thresholds. Data on asymptomatic spread inform key public health decisions, including lockdowns, travel restrictions, mask mandates, and the allocation of resources such as testing, hospital beds, and vaccines. Additionally, identifying asymptomatic cases supports more effective vaccine distribution and helps prevent redundancy in areas with high levels of natural immunity [37,38].

By integrating GICA testing alongside RT-PCR, healthcare providers can enhance diagnostic accuracy and obtain a more nuanced understanding of infection status, particularly in time-sensitive clinical environments. This approach not only aids in immediate decision-making but also contributes to broader infection control measures, especially during outbreak scenarios.

While the study offers valuable insights, it is not without limitations. The sample size, although sufficient to establish statistical significance, may not fully capture the variability that would be observed across a larger or multi-center population. Moreover, both the GICA and RT-PCR methods, despite being widely validated, are susceptible to technical variability influenced by operator handling, reagent quality, and specimen integrity. Future studies should aim to replicate these findings across broader cohorts and evaluate potential confounding factors such as co-infections, disease stage, and immune response variability that may influence test performance.

Further validation of the GICA method using nasopharyngeal swabs is planned, involving larger populations and multiple testing sites. Efforts will focus on improving the test’s sensitivity and negative predictive value through protocol optimization and improved reagent formulations. Additionally, a preliminary analysis using 63 saliva specimens demonstrated promising diagnostic potential. Interestingly, the inclusion of the detergent Triton X-100 was found to significantly enhance the preparation of saliva samples for GICA testing, highlighting the importance of optimizing sample processing protocols to improve assay reliability and minimize variability.

Future research should focus on refining and optimizing the GICA method to improve its diagnostic accuracy, consistency, and adaptability across different specimen types. Expanding validation efforts for both nasopharyngeal and saliva-based testing will be critical to assessing its performance in a range of clinical environments, including hospitals, nursing homes, mobile clinics, and community health settings. With further improvement, GICA could become a vital tool for the rapid detection and management of emerging infectious diseases, especially in resource-limited regions where accessible and timely diagnostics are essential. The continued development of such decentralized, low-cost diagnostic technologies could also play a pivotal role in strengthening global pandemic preparedness.

## 5. Conclusions

The findings indicate that a GICA (gold immunochromatographic assay) has the potential to be a valuable complementary diagnostic tool alongside RT-PCR, especially in resource-limited settings and scenarios requiring high-throughput screening. By incorporating a GICA into comprehensive testing strategies, healthcare systems can significantly bolster early detection efforts, ensuring timely identification of infected individuals. This integration can also enhance the accessibility of diagnostic services, making it easier for communities with limited resources to access reliable testing.

Moreover, the use of GICAs can alleviate the burden on RT-PCR testing facilities, allowing them to focus on cases where high sensitivity is paramount. This dual approach can optimize the allocation of resources, streamline testing processes, and ultimately strengthen the overall pandemic response. By leveraging the strengths of both GICAs and RT-PCR, healthcare systems can create a more resilient and adaptive framework for managing current and future viral outbreaks.

In summary, the strategic integration of GICAs into existing diagnostic protocols offers a promising avenue for improving public health outcomes, particularly in settings where rapid and accessible testing is crucial. This approach not only enhances early detection but also supports broader efforts to contain and mitigate the impact of pandemics.

## Figures and Tables

**Figure 1 diagnostics-15-01362-f001:**
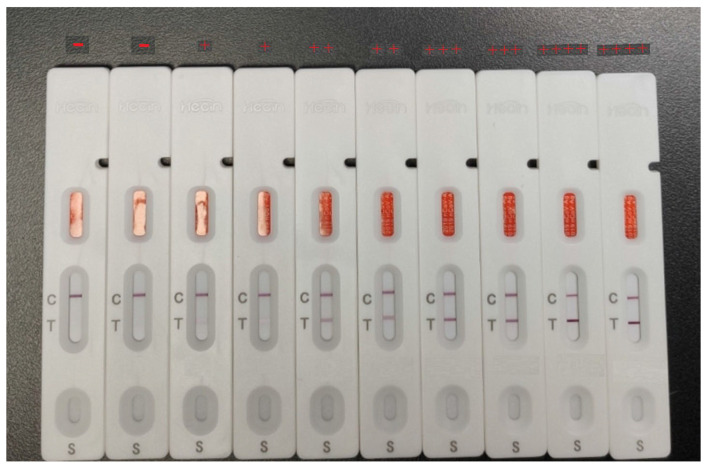
The intensity of the reaction line is used to class antigen levels into five distinct grades.

**Figure 2 diagnostics-15-01362-f002:**
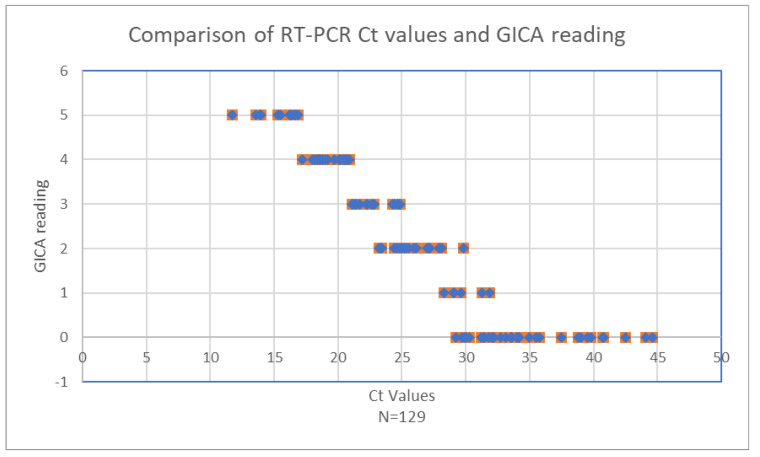
Comparison of RT-PCR Ct values and GICA readings.

**Figure 3 diagnostics-15-01362-f003:**
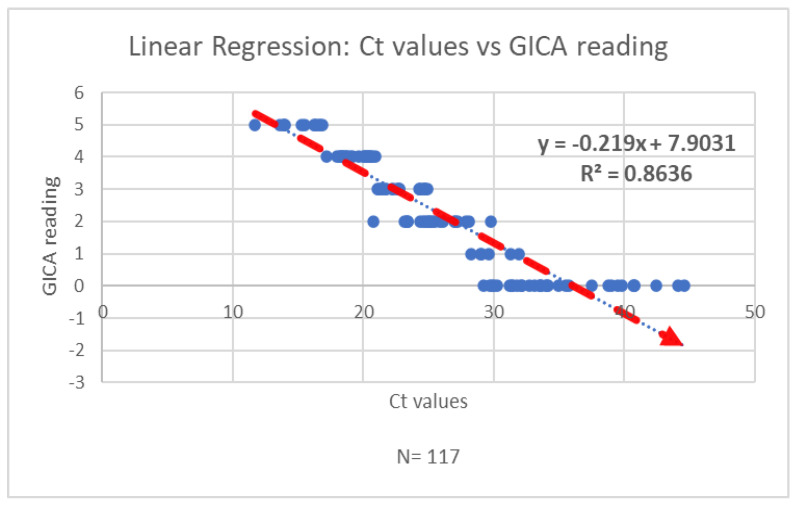
The regression analysis of Ct values and GICA readings indicates a strong negative correlation between the two variables.

**Figure 4 diagnostics-15-01362-f004:**
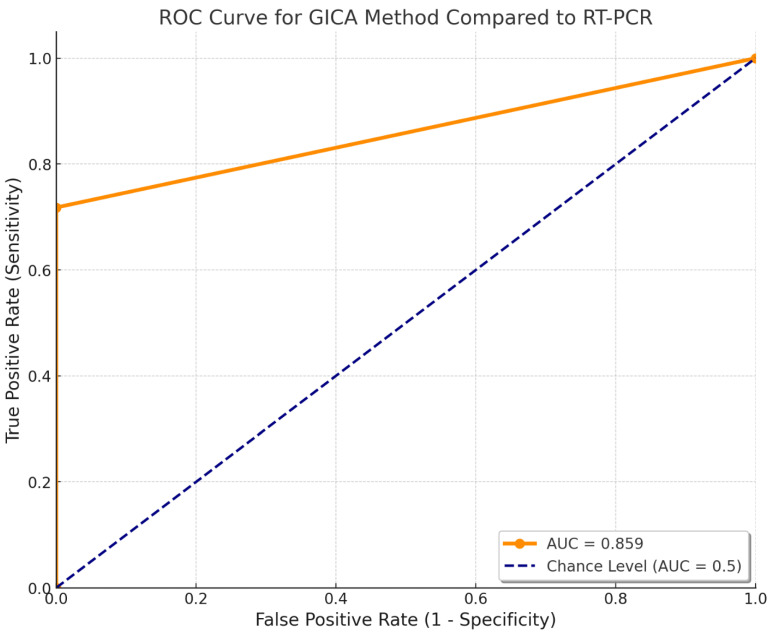
Receiver operating characteristic of the GICA method. The AUC was approximately 0.859.

**Table 1 diagnostics-15-01362-t001:** The intensity pattern of the test and control lines scanned by the Colloidal Gold Immunoassay Analyzer IVD-CG1 (Guangxi IVD-Biotechnology Co., Ltd., Nanning, Guangxi, P. R. China).

Sample Number	Test Result		Grade
1	C:2613.290	T:2.800	−
2	C:2635.149	T:3.208	−
3	C:2679.448	T:430.283	+
4	C:2508.893	T:372.090	+
5	C:2596.239	T:1185.282	++
6	C:2551.200	T:1082.041	++
7	C:2403.839	T:2521.277	+++
8	C:2568.612	T:2624.942	+++
9	C:2312.836	T:3312.668	++++
10	C:2382.953	T:3105.827	++++

C: Control line; T: Test line; Numerical value: Signal intensity.

**Table 2 diagnostics-15-01362-t002:** Comparison of RT-PCR and GICA test results.

	RT-PCR Positive	RT-PCR Negative	Total
**GICA positive**	84	0	**84**
**GICA negative**	33	12	**45**
**Total**	**117**	**12**	**129**

**Table 3 diagnostics-15-01362-t003:** The performance of the GICA testing.

Parameters	Values	95% CI
Sensitivity	71.8%	63.6% to 79.9%
Specificity	100%	100 to 100%
Positive Predictive Value (PPV)	100%	100% to 100%
Negative Predictive Value (NPV)	26.7%	13.7% to 39.6%
Accuracy	74.4%	66.9% to 81.9%

**Table 4 diagnostics-15-01362-t004:** Comparison of RT-PCR Ct values and GICA readings for detection of SARS-CoV-2.

RT-PCR Ct Value Range	GICA Reading	No. of Samples Positive for RT-PCR *	No. of Samples Positive for GICA	Positive Rate
**11.7–16.9**	5	13	13	100%
**17.2–20.9**	4	24	24	100%
**21.1–24.9**	3	28	28	100%
**25.0–28.1**	2	12	12	100%
**28.3–31.9**	1	16	7	56.30%
**32.1–44.6**	0	14	0	0%

* 10 RT-PCR positive specimens by the BioFire COVID-19 assay without a Ct value.

## Data Availability

The data presented in this study are available upon request from the corresponding authors.

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
