# Peer review of "Comparative Analysis of RT-PCR and a Colloidal Gold Immunochromatographic Assay for SARS-CoV-2 Detection"

_diagnostics, 2025, doi:10.3390/diagnostics15111362_

Round 1

Reviewer 1 Report

Comments and Suggestions for Authors

The article "Comparative Analysis of RT-PCR and Colloidal Gold Immunochromatographic Assay for SARS-CoV-2 Detection" correlates with the subject of the journal. The text of the introduction and conclusion is written quite clearly and in detail. However, a major revision is required for its acceptance for publication.

    The biggest question arises when reading lines 167-175. It is not entirely clear how to assign knowledge 2, 3, 4, and 5 to the results of immunochromatographic analysis. This is a rather subjective assessment. Authors are asked to provide references to studies in which the results of immunochromatographic analysis were assessed in a similar way. It is also necessary to confirm these results using computer programs for image evaluation, such as Image J.

To enhance the significance of the article, authors must also provide images of test strips after GICA. For example, illustrating the process of signal evaluation (lines 167-175).

Author Response

The article "Comparative Analysis of RT-PCR and Colloidal Gold Immunochromatographic Assay for SARS-CoV-2 Detection" correlates with the subject of the journal. The text of the introduction and conclusion is written quite clearly and in detail. However, a major revision is required for its acceptance for publication.

The biggest question arises when reading lines 167-175. It is not entirely clear how to assign knowledge 2, 3, 4, and 5 to the results of immunochromatographic analysis. This is a rather subjective assessment. Authors are asked to provide references to studies in which the results of immunochromatographic analysis were assessed in a similar way. It is also necessary to confirm these results using computer programs for image evaluation, such as Image J.

To enhance the significance of the article, authors must also provide images of test strips after GICA. For example, illustrating the process of signal evaluation (lines 167-175).

Thank you very much for taking the time to review this manuscript. Please find the detailed responses below and the corresponding revisions in track changes in the re-submitted files. 

We added references for quantification analysis.

  1. Zhang Y, Xiao W, Kong H, Cheng J, Yan X, Zhang M, Wang Q, Qu H, Zhao Y. A Highly Sensitive Immunochromatographic Strip Test for Rapid and Quantitative Detection of Saikosaponin d. Molecules. 2018 Feb 6;23(2):338. doi: 10.3390/molecules23020338. PMID: 29415494; PMCID: PMC6017486
  2. Zhang Y, Cao P, Lu F, Cheng J, Qu H. Development of a Lateral Flow Immunochromatographic Strip for Rapid and Quantitative Detection of Small Molecule Compounds. J Vis Exp. 2021 Nov 13;(177). doi: 10.3791/62754. PMID: 34842231
  3. Xu M, Lu F, Lyu C, Wu Q, Zhang J, Tian P, Xue L, Xu T, Wang D. Broad-range and effective detection of human noroviruses by colloidal gold immunochromatographic assay based on the shell domain of the major capsid protein. BMC Microbiol. 2021 Jan 11;21(1):22. doi: 10.1186/s12866-020-02084-z. PMID: 33430771; PMCID: PMC7798207
  4. Li X, Yin Y, Pang L, Xu S, Lu F, Xu D, Shen T. Colloidal gold immunochromatographic assay (GICA) is an effective screening method for identifying detectable anti-SARS-CoV-2 neutralizing antibodies. Int J Infect Dis. 2021 Jul;108:483-486. doi: 10.1016/j.ijid.2021.05.080. Epub 2021 Jun 6. PMID: 34091005; PMCID: PMC8180344.

In our case Colloidal Gold Immunoassay Analyzer by Guanxi IVD-Biotechnology Co., Ltd. was used for analysis. The images of the results together with the corresponding quantification analysis are included in the revised manuscript (Figure 1 and Table 1).

Reviewer 2 Report

Comments and Suggestions for Authors

Type of manuscript: Article
Title: Comparative Analysis of RT-PCR and Colloidal Gold Immunochromatographic Assay for SARS-CoV-2 Detection

Dear editorial office

It is my pleasure to read and review this above titled paper for journal “Diagnostics”.

In total, there is a good and well-written paper but needs minor revision before final decision.

I feel the idea presented in this paper can be useful for next unfortune pandemic hopefully never occured!

  • High specificity (100%) and PPV (100%) for GICA underscore its utility in confirming positive cases. However, I am wondering if authors can provide a better discussion about it in revised version.
  • Small Sample: its generalizability is under question mark. I feel it needs a rewrite in discussion. Try to compensate it.
  • Single-Center, Retrospective Design: Some bias may be exists, please point it in limitations. I know this research is finished but it can be pointed as a new suggestion for upcoming studies.
  • Unvaccinated Cohort: this part of samples may cause cofounding impact on final conclusion. I think the challenge if asymptomatic carriers is serious and it should be repeated in new text, so I suggest a paper in this field for covering the idea. I expected to read new text in discussion part.

*******Challenges of managing the asymptomatic carriers of SARS-CoV-2. Travel medicine and infectious disease37, p.101677.

***** Tan C, Xiao Y, Meng X, Huang X, Li C, Wu A. Asymptomatic SARS-CoV-2 infections: What do we need to know? Infect Control Hosp Epidemiol. 2021 Jan;42(1):114-115. doi: 10.1017/ice.2020.201. Epub 2020 May 6. PMID: 32372742; PMCID: PMC7242771.

Comments on the Quality of English Language

good

Author Response

Dear editorial office

It is my pleasure to read and review this above titled paper for journal “Diagnostics”.

In total, there is a good and well-written paper but needs minor revision before final decision.

I feel the idea presented in this paper can be useful for next unfortune pandemic hopefully never occured!

Thank you very much for taking the time to review this manuscript. Please find the detailed responses below and the corresponding revisions in track changes in the re-submitted files. 

  • High specificity (100%) and PPV (100%) for GICA underscore its utility in confirming positive cases. However, I am wondering if authors can provide a better discussion about it in revised version.

We revised the discussion (see lines 307 – 314).

  • Small Sample: its generalizability is under question mark. I feel it needs a rewrite in discussion. Try to compensate it.

Please note that our clinic laboratory was one of multiple sites for validation of this GICA assay. Our report only includes the data generated in our own laboratory. It does not include all the data generated from other laboratories. This assay is approved by China FDA and CE. Evaluation studies were performed in multiple clinical laboratories in China, Greece, Austria, United States, France, British and Germany, etc.

  • Single-Center, Retrospective Design: Some bias may be exists, please point it in limitations. I know this research is finished but it can be pointed as a new suggestion for upcoming studies.

As noted above, our clinic laboratory was one of multiple sites for validation of this GICA assay.

  • Unvaccinated Cohort: this part of samples may cause cofounding impact on final conclusion. I think the challenge if asymptomatic carriers is serious and it should be repeated in new text, so I suggest a paper in this field for covering the idea. I expected to read new text in discussion part.
  1. Challenges of managing the asymptomatic carriers of SARS-CoV-2. Travel medicine and infectious disease37, p.101677.
  2. Tan C, Xiao Y, Meng X, Huang X, Li C, Wu A. Asymptomatic SARS-CoV-2 infections: What do we need to know? Infect Control Hosp Epidemiol. 2021 Jan;42(1):114-115. doi: 10.1017/ice.2020.201. Epub 2020 May 6. PMID: 32372742; PMCID: PMC7242771.

This study is intended to assess GICA utility among a population with a similar background of unvaccinated status. The criteria for patient inclusion and exclusion were not based on symptomology. See line 125, sample collection, Materia and methods. Nevertheless, we appreciate your suggestions regarding asymptomatic patients since the identification of asymptomatic carriers by GICA is important for controlling pandemics. See discussion and references in Line 329 – 341.

Round 2

Reviewer 1 Report

Comments and Suggestions for Authors

The revision is OK